# Changes in Ultrasound Parameters of the Median Nerve at Different Positions of the Radiocarpal Joint in Patients with Carpal Tunnel Syndrome

**DOI:** 10.3390/s24144487

**Published:** 2024-07-11

**Authors:** Tomasz Wolny, Katarzyna Glibov, Michał Wieczorek, Rafał Gnat, Paweł Linek

**Affiliations:** 1Musculoskeletal Elastography and Ultrasonography Laboratory, Institute of Physiotherapy and Health Sciences, The Jerzy Kukuczka Academy of Physical Education, 40-065 Katowice, Poland; r.gnat@awf.katowice.com (R.G.); linek.fizjoterapia@vp.pl (P.L.); 2Department of Internal Diseases, Rehabilitation and Physical Medicine, Military Medical Academy, Memorial Teaching H of The Medical University of Lodz—Central Veterans Hospital, 90-419 Lodz, Poland; katarzyna.glibov@umed.lodz.pl; 3Department of Neurological Rehabilitation, The Health Center in Mikołów Ltd., 43-190 Mikołów, Poland; limmeusz@gmail.com

**Keywords:** ultrasound imaging, shear wave elastography, carpal tunnel syndrome, entrapment neuropathy

## Abstract

Ultrasound imaging (US) is being increasingly used to aid in the diagnosis of entrapment neuropathies. This study aims to evaluate the shear modulus and cross-sectional area (CSA) of the median nerve in patients with carpal tunnel syndrome (CTS). A total of 35 patients with CTS participated in the study. CSA and shear modulus testing were performed in shear wave elastography (SWE) mode in five positions of the right and left radiocarpal joints (intermediate position 0°, 45° of extension, maximum extension, 45° of flexion, and maximum flexion). There were significant side-to-side differences in the median nerve shear modulus at each wrist position as compared to the asymptomatic side. There were significant side-to-side differences in the median nerve CSA at each wrist position as compared to the asymptomatic side. Shear modulus increases in patients with CTS at different angular positions of flexion and extension of the radiocarpal joint. In individuals with CTS, the CSA of the median nerve is greater on the symptomatic side compared to the asymptomatic side. The CSA decreases in positions of maximum extension and 45° of flexion and in maximum flexion relative to the resting position.

## 1. Introduction

Carpal tunnel syndrome (CTS) is the most common peripheral neuropathy of the upper limb, characterized by multiple sensory and motor symptoms [1,2]. CTS is caused by compression of the median nerve in the carpal tunnel and is, therefore, often referred to as entrapment syndrome [3]. Due to its significant prevalence (3.8–5.8%) [4,5], progressively worsening symptoms that initially involve sensory impairment of the hand and, over time, lead to impaired motor function of the hand, CTS has a significant impact on working life, activities of daily living, and a perception of overall quality of health [6,7,8]. This, in turn, has important socio-economic implications, especially as CTS often affects people of working age, resulting in higher absenteeism and lower productivity at work [9]. It is, therefore, important to make a prompt and accurate diagnosis and undertake effective treatment.

The diagnosis of CTS is usually based on clinical history and symptoms (nocturnal paresthesia, numbness, and tingling in the area of the median nerve, pain), positive functional testing (Phalen test, Tinel sign), and electrodiagnostic testing (nerve conduction study—NCS) [10,11]. Some authors indicated that the symptoms reported by patients and the clinical examination are sufficient to make a diagnosis of CTS and that NCS testing is only useful in confirming the diagnosis and assessing the severity of CTS [12]. Other authors consider the NCS test to be the gold standard for the diagnosis of CTS [13]. Proponents of NCS-based diagnosis of CTS highlight its high sensitivity and specificity of 80–92% and 80–99%, respectively [14,15]. However, some authors question the value of the NCS test in the diagnosis of CTS, as it has been shown that 16–34% of people with clinically diagnosed CTS had normal NCS parameters [16,17,18]. Therefore, the lack of a gold standard for examinations and confirmation of the diagnosis of CTS is increasingly emphasized, and the increasing potential of medical ultrasonography (US) in the diagnosis of this neuropathy is being indicated [19,20].

US of the median nerve is gaining popularity as a diagnostic tool that is useful in the assessment of CTS [21]. It has the advantage of being highly available, accurate, minimally invasive, inexpensive, and accepted by patients [11,22]. US allows a visual assessment of the median nerve and surrounding structures and an evaluation of its dynamic mobility [22]. US also allows the diagnostician to assess the surrounding tissues for secondary causes of CTS, such as tumors, cysts, and hematomas, that cannot be diagnosed with NCS [10]. In the US examination of the median nerve, cross-sectional area (CSA), flatting ratio, palmar bowing of the flexor retinaculum, nerve mobility, changes in nerve echogenicity, and vascularity are assessed [10,23]. The usefulness of strain elestography [24] and shear wave elastography (SWE) in the diagnosis of CTS median nerve elasticity has also been demonstrated [25,26].

Of the aforementioned parameters, median nerve CSA at the level of the tunnel inlet is commonly used as it shows the highest sensitivity and specificity in the diagnosis of CTS [27,28]. SWE of the median nerve is also characterized by high sensitivity and specificity in the diagnosis of CTS [26]. In studies of people with CTS, a characteristic feature is the analysis of the US of the median nerve in the neutral position of the radiocarpal joint [11,22,29]. However, it seems reasonable to assess the US of the median nerve in patients with CTS in different radiocarpal joint positions because the change in tension and tissue relationship within the carpal tunnel leads to a change in tunnel pressure, which may have a significant impact on the median nerve [30,31]. Anatomical studies have demonstrated that the pressure in the carpal tunnel is lowest in the neutral position and increases significantly when the wrist moves into flexion or extension [30,31]. Therefore, two tests (Phalen and reverse Phalen) are used in clinical diagnosis, where the occurrence of symptoms is assessed in extreme wrist flexion and extension [32,33]. Brininger et al. [34] found that the use of a wrist splint holding the wrist in a neutral position (0°) produced a more favorable therapeutic effect than the use of a splint in a 20° extension position. The authors explained that, outside the neutral position, the lumbrical muscles move into the carpal tunnel, thus increasing the pressure in the tunnel, which has a detrimental effect on the nerve [34]. Studies of the ulnar nerve in different angular positions of the elbow joint showed that the CSA parameter did not change significantly, but nerve stiffness, as measured by SWE, increased significantly with the degree of elbow joint flexion [30]. Furthermore, the increase in nerve stiffness was significantly greater in the limb with symptoms of cubital tunnel syndrome relative to the asymptomatic limb [30]. Thus, we hypothesize that in different radiocarpal joint positions, the change in pressure and stress on the tissues surrounding the nerve affect the US parameters of the median nerve. Thus, it may be that the US examination of the median nerve outside the neutral position shows more prominent differences in people with CTS. Therefore, this study aims to evaluate the basic US parameters (CSA and SWE) in patients with unilateral CTS at different angular positions of the radiocarpal joint. This information may improve the understanding of median nerve lesions in people with CTS and thus expand the role of US in the diagnosis of CTS.

## 2. Materials and Methods

### 2.1. Study Design

This study was an observational study with repeated US measurements of the median nerve at different angular positions of the radiocarpal joint. The examinations were conducted in a medical outpatient clinic in southern Poland and the Musculoskeletal Elastography and Ultrasonography Laboratory (the Jerzy Kukuczka Academy of Physical Education). Participants were people with clinically and electrophysiologically diagnosed CTS who were referred by their doctor for physiotherapy treatments. If a person agreed to participate in the study and met the inclusion criteria, they were eligible for further examinations. Each individual who met the inclusion criteria had a US examination performed. All US examinations were performed by a physiotherapist with more than 10 years of experience in US of the peripheral nerves. The examination included imaging the median nerve at the level of the carpal tunnel inlet, followed by measurement of the CSA and shear modulus of the median nerve in 5 positions of the radiocarpal joint (neutral position, 45° of extension, maximum extension, 45° of flexion, and maximum flexion). The person performing the US measurements was blinded to the study aim and the patient’s condition. Prior to the study, participants were asked not to give any information to the investigator about their health status. All participants were informed about the study. Written informed consent was obtained from all participants. All study procedures were performed according to the Declaration of Helsinki of 1975 and revised in 1983. The study was approved by the Bioethics Committee for Scientific Research of the Jerzy Kukuczka Academy of Physical Education in Katowice (No. 8/2019).

### 2.2. Participants

Each subsequent person who presented to the outpatient clinic with clinically and electrophysiologically diagnosed CTS and met specific criteria was invited to participate in the study. Inclusion criteria for the examinations were at least two positive clinical symptoms of CTS (night-time paraesthesia, numbness and tingling in the area of the median nerve, positive Phalen’s test, positive Tinel’s sign, pain in the wrist area radiating to the shoulder) and abnormalities in the median nerve conduction (diminished nerve conduction values below 50 m/s, increased motor latency above 4 m/s) confirmed by NCS. Exclusion criteria were lack of consent to participate in the study, previous conservative or surgical treatment of CTS, cervical radiculopathy, tendinitis, rheumatoid disease, diabetes mellitus, pregnancy, thenar muscle atrophy, history of wrist injury, and other peripheral neuropathies of the upper limbs.

### 2.3. Ultrasound Measurements

US measurements were performed with a Hologic Supersonic Mach 30 ultrasound scanner (Supersonic Imagine, Aix En Provence, France) using a linear transducer array (5–18 MHz; Super Linear SL18-5, Supersonic Imagine).

Before the test, a piece of metal plate was taped to the back of the hand, to which a digital inclinometer with a built-in magnet was attached. This allowed continuous monitoring of the angular alignment of the radiocarpal joint during US examination (Figure 1).

During the examination, the participant was in a sitting position, the arm was positioned along the torso, the elbow joint was flexed to 90°, and the forearm was supinated and resting on a height-adjustable therapy couch, allowing the position to be adjusted to the participant’s body height. The participant’s hand was outside the couch so that the examination could be easily performed, and the position of the radiocarpal joint could be controlled with an inclinometer. During the examination, the patient was required to actively maintain a previously set body position. In this way (actively maintaining body posture), the measurement result can be close to the carpal tunnel pressure that occurs during activities of daily living, which are mostly carried out vigorously. Only the weight of the inclinometer might slightly affect the measurements. However, its weight was small (60 g). Three measurements were taken in each position. A 30 s rest was administered after every 3 measurements in a specific joint position. The measurement was always taken at the same place (at the level of the carpal tunnel inlet), which was marked on the skin with a marker.

US images were collected using SWE mode, which was installed on the US machine. Images were collected in 5 different positions of both radiocarpal joints (intermediate position 0°, 45° of extension, maximum extension, 45° of flexion, and maximum flexion). To control and minimize the pressure of the head on the skin, a correspondingly large amount of hydrogel was applied to the area examined. After imaging the median nerve at the level of the carpal tunnel inlet, transverse scans were collected at each angular position (controlled with an inclinometer), starting each time with a neutral position (0°), followed by 45° of extension, maximum extension, 45° of flexion, and maximum flexion of the radiocarpal joint. Three images and then three measurements were taken at each angular position, and the average values from these measurements were used for the final analysis. For both CSA and shear modulus measurements, a contour was made of the median nerve at the inner hypoechoic border using a Q-Box Trace option implemented in the US machine (Figure 2). Both parameters were automatically calculated in the US machine and expressed in square millimeters and kilopascals (kPa) for CSA and shear modulus, respectively. CSA is related to nerve surface area, whereas shear modulus refers to the stiffness of the nerve. Higher values of shear modulus indicate higher nerve stiffness and vice versa. Detailed information about the basic physics and musculoskeletal applications of SWE is placed elsewhere [35].

### 2.4. Statistical Analyses

Data were analyzed using STATISTICA 13.3 PL (Statsoft, Tulsa, OK, USA) software. To compare the symptomatic and asymptomatic sides, a one-way ANOVA for repeated measurement was used. For significant main effects in the ANOVA, the planned comparisons were performed. To diminish a family-wise error rate for multiple comparisons, the Holms correction was implemented [36].

The results are presented in the Figures as a mean value and 95% confidence interval (CI) of the mean value. A significant difference was presented in the text as a mean difference with 95% CI. For all analyses, the threshold of the *p*-value considered as significant was set at ≤0.05 with additional Holm’s correction for multiple comparisons.

Additionally, an intraclass correlation coefficient (ICC) was calculated to assess intrarater reliability (ICC3,k) using three consecutive ultrasound measurements from both sides of all patients. Next, the standard error of measurement (SD × (1 − ICC)) and the smallest detectable differences (SDD) (SDD = 1.96 × SEM × 2) were also calculated in order to consider the clinical significance of the detected between-side differences.

## 3. Results

### 3.1. Participants

In total, 43 people were recruited, but 8 people did not meet the inclusion criteria for the study (4 people had previous conservative treatment, 1 person was diagnosed with cervical radiculopathy, 2 people suffered from diabetes, and 1 person reported a history of wrist injuries). A total of 35 Polish patients (from the Silesian Region) with unilateral CTS were included in the final analysis. In all study participants, CTS occurred in the dominant hand. The basic characteristics of the study group are presented in Table 1. All study participants reported nighttime paraesthesia and numbness and tingling in the area of the median nerve (35; 100%), positive Phalen’s test occurred in 26 (74%) subjects, positive Tinel’s sign occurred in 20 (57%) subjects, pain in the wrist area radiating to the shoulder occurred in 21 (60%) subjects.

### 3.2. US Measurement Reliability

During ICC calculation, the results from the symptomatic and asymptomatic extremities were pooled together. In all tested positions, the ICCs indicated excellent reliability of both shear modulus and CSA measurements (Table 2).

### 3.3. Shear Modulus

With regards to nerve median elasticity, there were side-to-side differences at each wrist position (*p* < 0.0001). Compared to the asymptomatic side, the mean shear modulus value of the median nerve in the symptomatic side was higher by 31.1 kPa (95% CI 24.6–37.6), 84.7 kPa (95% CI 60.2–109.2), 169.4 kPa (95% CI 149.4–189.4), 55.0 kPa (95% CI 49.5–60.5), and 201.3 kPa (95% CI 193.1–209.5) in the neutral wrist, 45° wrist extension, maximum wrist extension, 45° wrist flexion, and maximum wrist flexion, respectively (Figure 3). All the differences calculated were higher than the SDD values (Table 2).

### 3.4. CSA

With regards to nerve median CSA, there was a side-to-side difference at each wrist position (*p* < 0.0001). Compared to the asymptomatic side, the CSA of the median nerve in the symptomatic side was higher by 3.82 mm^2^ (95% CI 3.39–4.25), 3.73 mm^2^ (95% CI 3.32–4.14), 3.82 mm^2^ (95% CI 3.41–4.23), 3.51 mm^2^ (95% CI 3.04–3.98), and 3.73 mm^2^ (95% CI 3.31–4.14) in the neutral wrist, 45° wrist extension, maximum wrist extension, 45° wrist flexion, and maximum wrist flexion, respectively (Figure 4). All the differences calculated were higher than the SDD values (Table 2).

## 4. Discussion

The aim of this study was to evaluate the shear modulus and CSA of the median nerve in the carpal tunnel at different angular positions of the radiocarpal joint in patients with CTS. To the best of our knowledge, studies to date have not evaluated whether and how shear modulus and CSA of the median nerve change in CTS patients depending on the position of the radiocarpal joint. The results showed significant differences in shear modulus and CSA of the median nerve in the symptomatic as compared to the asymptomatic limb. The symptomatic side showed significantly higher shear modulus and CSA of the median nerve in each angular position of the radiocarpal joint compared to the asymptomatic side. It should be emphasized that shear modulus gradually increases with increasing angles of extension and flexion of the radiocarpal joint in both the symptomatic and asymptomatic limbs, but this increase is much greater in the symptomatic limb. The CSA of the median nerve significantly decreases with increasing angle of extension and flexion of the radiocarpal joint in both the symptomatic and asymptomatic limb and reaches its lowest values in the maximal positions of the joint.

SWE in the assessment of median nerve stiffness in patients with CTS has already been studied in several papers [25,26]. Kantarci et al. [25] assessed median nerve stiffness at the level of the carpal tunnel inlet in 37 patients with CTS and 18 healthy subjects. In these studies, the mean stiffness of the median nerve in patients with CTS was significantly higher (66.7 kPa) compared to healthy individuals (32.0 kPa) and higher in those with severe and extreme compared to mild and moderate CTS. They also showed high sensitivity and specificity (93.3% and 88.9%, respectively) and excellent agreement between investigators [25]. Nam et al. [26] examined median nerve stiffness in 27 hands with CTS and 20 healthy hands at the level of the proximal carpal row and proximal to the pronator quadratus muscle. They also showed significantly higher median nerve stiffness (101.3 and 73.4 kPa, respectively) in people with CTS compared to healthy individuals (56.4 and 49.1 kPa, respectively). Mohammadi et al. [31] assessed median nerve stiffness in 70 wrists of patients with CTS and 54 wrists of healthy individuals at the level of the carpal tunnel inlet. These examinations showed that median nerve stiffness in people with CTS was significantly higher (56.55 kPa) compared to healthy individuals (23.71 kPa). Sernik et al. [37] also documented significantly higher median nerve stiffness in patients with CTS compared to healthy individuals, but the shear modulus values were significantly lower than in previous studies (average 19.4 kPa in people with CTS and 11.1 kPa in healthy individuals). Our study showed similar values to those presented in the study by Kantarci et al. [25] and Mohammadi et al. [31] (69 kPa in patients with CTS and 37.8 kPa in healthy individuals). It should be stressed, however, that in all the studies presented here, the differences between individuals (wrists) with diagnosed CTS and healthy/asymptomatic individuals (wrists) were significant and significantly greater in the symptomatic limbs. It is likely that the differences in median nerve stiffness are caused by the severity of CTS, the different locations of the shear modulus testing, and other differences in testing methodologies.

Most studies to date have assessed median nerve stiffness in the intermediate position of the radiocarpal joint [25,26]. Mohammadi et al. [31] assessed median nerve stiffness in a radiocarpal joint position of 30–45° of extension. In our study, we assessed median nerve stiffness in five positions of the radiocarpal joint (at rest, 45° of extension, at maximum extension, 45° of flexion, and maximum flexion), which is new to the SWE examinations with CTS. Our study showed that changing the position of the joint significantly increases median nerve stiffness at the level of the carpal tunnel inlet in both the symptomatic and asymptomatic hand. However, it should be noted that the increase in median nerve stiffness in symptomatic hands was significantly greater compared to asymptomatic hands. The increased stiffness of the median nerve in symptomatic compared to asymptomatic hands at neutral radiocarpal joint position can be explained by increased pressure in the carpal tunnel in patients with CTS [33], which disturbs circulation in the nerve, causing swelling, which in the long-term leads to fibroblast invasion, scarring, and fibrosis of the nerve, resulting in increased stiffness [25]. A further significant increase in nerve stiffness when increasing the degree of extension and flexion of the radiocarpal joint indicates that other mechanisms also have an effect. It has been shown that as the degree of flexion and extension of the radiocarpal joint increases, its volume decreases [38,39], which may have the effect of increasing the pressure and, therefore, the stiffness of the nerve in the carpal tunnel. This can explain the increase in nerve stiffness not only in symptomatic but also in asymptomatic limbs. Another mechanism for the increase in median nerve stiffness during the extension of the radiocarpal joint may be the gradual stretching of the nerve, which increases its tension by 9.6% [40]. If the nerve compression occurring in CTS further restricts its longitudinal and transverse mobility, as confirmed by numerous studies [41,42,43], the increase in tension on the median nerve and the consequent increase in its stiffness may be even greater. The results of this study show significant changes in median nerve stiffness in the carpal tunnel in patients with CTS and indicate that nerve stiffness should be tested at different angular positions. Our finding has another very important practical implication regarding the prevention and treatment of CTS, indicating that the nerve is least stressed in the intermediate position of the radiocarpal joint, and therefore, this joint position should be preferred in the prevention and treatment of CTS.

The usefulness of CSA assessment in the diagnosis of CTS is frequently highlighted in many scientific reports [10,12,19,22]. Increased CSA in patients with CTS proximal to the site of compression was one of the first findings [10]. A meta-analysis based on 28 studies (3995 radiocarpal joints with CTS) showed that increased CSA is the best ultrasound diagnostic criterion in patients with CTS, characterized by high sensitivity and specificity (87% and 83%, respectively) [44]. Assessment of the CSA at the inlet of the carpal tunnel has been shown to be not only an important but also an easy parameter to assess in routine clinical examinations to confirm CTS [12]. Ahmed et al. [12] showed a strong correlation between increased median nerve CSA and CTS severity based on NCS.

Similar to the assessment of SWE of the median nerve, the effect of the angular position of the radiocarpal joint on CSA in patients with CTS has not been studied. In most studies to date, assessment of the median nerve has been performed in the resting position of the radiocarpal joint [22,25,26]. Previous studies indicate that normal median nerve CSA values are <10 mm^2^ [44,45], with values between 10 and 12.9 mm^2^ indicating mild CTS, 13–14.9 mm^2^ indicating moderate CTS, and >15 mm^2^ indicating severe CTS [46]. Our studies in the resting position of the radiocarpal joint found similar values (the mean CSA was 9.3 mm^2^ in the asymptomatic limb and 13.1 mm^2^ in the symptomatic limb). Interestingly, changing the angle of the radiocarpal joint has an effect on the CSA of the median nerve in both the symptomatic and asymptomatic limb. In the symptomatic limb at rest and at 45° of extension, the CSA did not differ. At maximum extension, it was smaller by an average of 0.7 mm^2^, by 1 mm^2^ at 45° of flexion, and by 1.5 mm^2^ at maximum flexion compared to the resting position. In the asymptomatic limb, the CSA of the nerve was smaller by 0.8 mm^2^ in maximum extension, by 0.7 mm^2^ in 45° of flexion, and by 1.5 mm^2^ in maximum flexion compared to the resting position. It should be noted that the difference between the resting position and the position of maximum flexion and extension was statistically significant. It is likely that the change in CSA of the median nerve in maximal extension will be influenced by the lengthening of the nerve [40] and, in progressive flexion of the radiocarpal joint, by a decrease in the volume of the carpal tunnel [38,39] and the compression of the structures within the tunnel on the median nerve. Furthermore, it should be emphasized that all studies to date have indicated that a significant increase in radial nerve CSA and a cut-off value of >10 mm^2^ are of the greatest importance in the diagnosis of CTS, which was partly confirmed by the current study, but that decreasing median nerve CSA values, which may result in worsening CTS symptoms in extreme joint positions, should be taken into account in the diagnosis of CTS and in prevention and therapy planning.

Recently, reports on the possibility of using US and SWE in the diagnosis of CTS have increasingly appeared in the literature. El-Maghraby et al. [47] assessed the usefulness of the combined diagnostic utility of B-mode US and SWE for assessing the severity of CTS in comparison to electrodiagnostic tests. Based on the research results, the authors concluded that the integration of US, SWE, and electrodiagnostic tests provides a comprehensive approach to evaluating anatomical and neurological changes and guides management decisions for CTS. Similar conclusions were drawn from their research by Martikkala et al. [48], indicating that shear wave velocity in the median nerve at the carpal tunnel inlet increases in CTS and correlates to the neurophysiological CTS severity equivalently to CSA measured at the same site. In turn, Prakash et al. [49] assessed the diagnostic effectiveness of high-frequency US and SWE in grading the severity of CTS. The authors conclude that the combined utility of US and SWE may serve as a painless and cost-effective alternative to nerve conduction study in grading the severity of CTS. Interesting research was conducted by Khademi et al. [50] assessing SWE and CSA of the median nerve in people with CTS after using neurodynamic techniques. Researchers have shown that one session of neurodynamic techniques reduced the stiffness and CSA of the median nerve at the wrist in patients with CTS. The authors emphasize that this confirms that SWE is able to quickly detect the immediate biomechanical changes of the median nerve. Wu et al. [51] showed the usefulness of conventional US and SWE in the clinical assessment of a patient’s nerve recovery after surgery. We believe that performing such an examination in different angular positions may better reflect changes in the morphology of the nerve. Our report expands the topic of the use of US and SWE in the diagnosis of CTS, indicating that it is beneficial to examine the median nerve (CSA and shear modulus) not only in the neutral position of the wrist but also in other angular positions. This will be important not only for more detailed diagnostics but also for a better assessment of the final results of the therapy used.

A novelty and a great value of this study is the demonstration that both ultrasound parameters (SWE and CSA) depend on the angle of the radiocarpal joint. It was shown that there are significant differences in median nerve stiffness and CSA between the symptomatic and asymptomatic limbs and that changing the joint angle also significantly affects both ultrasound parameters. We believe that this finding will bring important information to the diagnosis, prevention, treatment, and evaluation of its effectiveness in patients with CTS.

Our study has some limitations. The first limitation is the relatively small group of CTS patients studied, which specifically focused on one type of patient (Polish). This does not allow us to make a generalized approach to the general population. The second limitation is that the nerve conduction examinations were performed only in the symptomatic limb and not in the asymptomatic limb. Consequently, the presence of nerve conduction disorders cannot be ruled out even before the CTS symptoms. On the other hand, the subjects did not report any subjective symptoms of ulnar nerve neuropathy. Such a study would also be unethical because of its burdensome nature. Another limitation is the weight of the inclinometer, which, when attached to the patient’s hand, could have affected the tension of the tissues and thus the measurement results, although the inclinometer used in the study has a relatively low weight (60 g).

## 5. Conclusions

The median nerve in patients with CTS is characterized by greater stiffness on the symptomatic compared to the asymptomatic side at different angular positions of the radiocarpal joint. Median nerve stiffness increases with increasing degree of flexion and extension of the radiocarpal joint in both the symptomatic and asymptomatic limbs. The median nerve in patients with CTS is characterized by a larger CSA on the symptomatic as compared with the asymptomatic side. The CSA of the median nerve decreases in positions of maximum extension and 45° of flexion and in maximum flexion relative to the resting position. US examination (assessment of shear modulus and CSA) of the median nerve at different angular positions of the radiocarpal joint can be helpful in the diagnosis and monitoring of lesions in patients with CTS. Further research in this area is warranted and necessary.

## Figures and Tables

**Figure 1 sensors-24-04487-f001:**
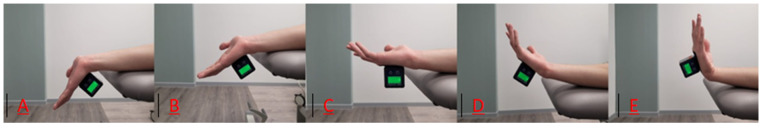
Hand position and angle measurement for ultrasound examination ((**A**)—maximum extension, (**B**)—45° extension, (**C**)—neutral position, (**D**)—45° flexion, (**E**)—maximum flexion).

**Figure 2 sensors-24-04487-f002:**
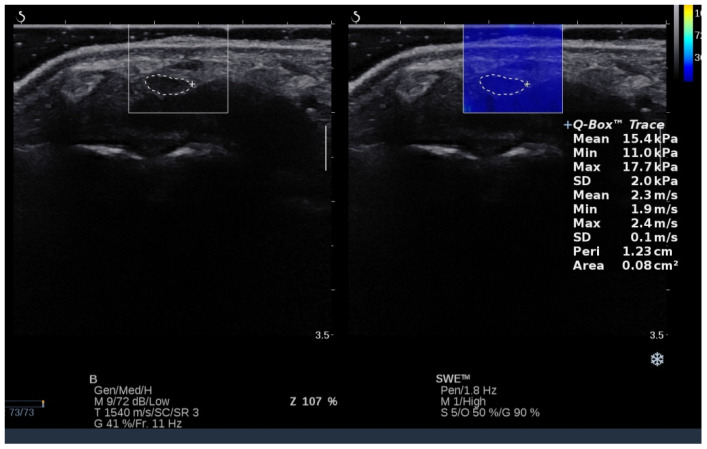
Median nerve localization in shear wave elastography mode and measurement example of CSA and shear modulus.

**Figure 3 sensors-24-04487-f003:**
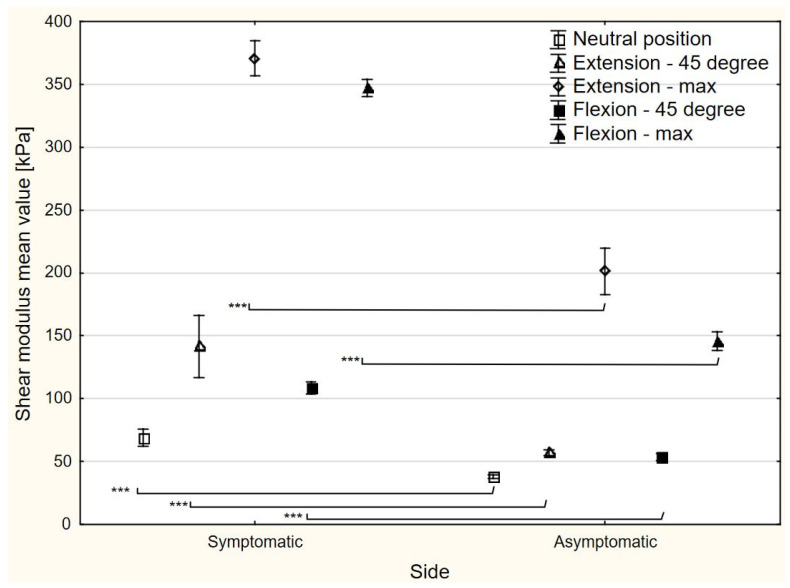
The shear modulus changes in different positions of the radiocarpal joint. *** statistically significant difference.

**Figure 4 sensors-24-04487-f004:**
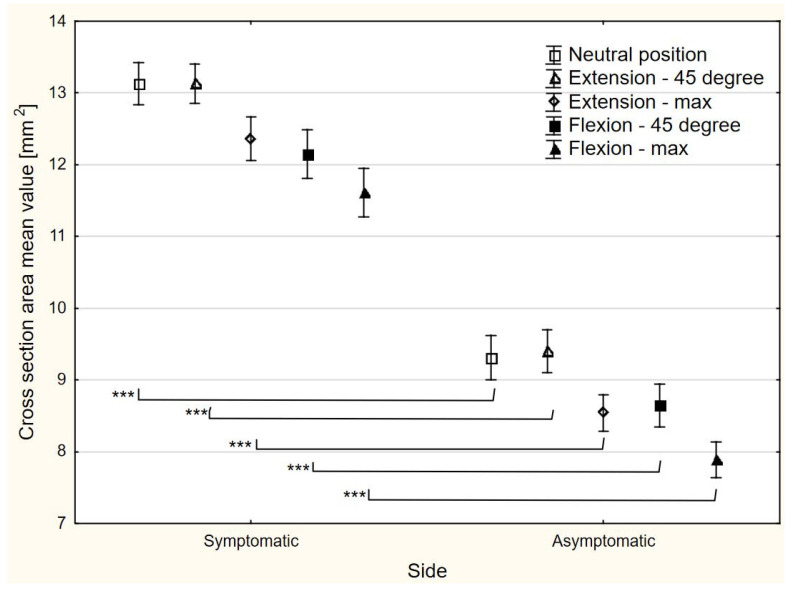
The cross-sectional area changes in different positions of the radiocarpal joint. *** statistically significant difference.

**Table 1 sensors-24-04487-t001:** Basic characteristics of the study group.

Characteristics of Study Participants	
Age (years)	55.4 (9.01; 37–72)
Height (cm)	165.6 (5.97; 155–180)
Body mass (kg)	71.1 (10.4; 50–95)
Gender (numbers)	26 (74%) female 9 (26%) male
Right-handed	31 (88%)
Left-handed	4 (12%)
NCS SCV (m/s)	23.4 (14.3; 0–44)
NCS ML	5.62 (0.87; 4.5–7.4)
NPRS	5.7 (1.51; 3–9)
BCTQ	
SSS	3.01 (0.66; 1.72–4.63)
FSS	2.23 (9.01; 1.12–4)
Duration of symptoms (month)	18 (4.25; 12–24)

NCS—Nerve Conduction Study; SCV—Sensory Conduction Velocity; ML—Motor latency; NPRS—Numerical Pain Rating Scale; BCTQ—Boston Carpal Tunnel Questionnaire; SSS—Symptomatic Severity Scale; FSS—Functional Status Scale.

**Table 2 sensors-24-04487-t002:** Intraclass Correlation Coefficient (ICC), standard error of measurement (SEM), and the smallest detectable differences (SDD) for all outcomes measured. The ICC was calculated based on three consecutive measurements from all patients and both hands.

Ultrasound Measurements		ICC_3,3_	SEM	SDD
Shear modulus	Neutral wrist position	0.99	2.16	6.00
	45° wrist extension	0.99	6.83	18.9
	Maximal wrist extension	0.99	9.99	27.7
	45° wrist flexion	0.99	3.05	8.46
	Maximum wrist flexion	0.99	10.3	28.5
CSA	Neutral wrist position	0.99	0.22	0.60
	45° wrist extension	0.99	0.21	0.58
	Maximal wrist extension	0.99	0.21	0.59
	45° wrist flexion	0.99	0.20	0.56
	Maximum wrist flexion	0.99	0.21	0.58

## Data Availability

Data are available on request.

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
