# Peer review of "Changes in Ultrasound Parameters of the Median Nerve at Different Positions of the Radiocarpal Joint in Patients with Carpal Tunnel Syndrome"

_sensors, 2024, doi:10.3390/s24144487_

Round 1

Reviewer 1 Report

Comments and Suggestions for Authors

The written of this manuscript is quite well. However, from scientific point of view, the description of the technical methods is not very clear. The manuscript needs to describe how the ultrasound shear wave was measured from ultrasonic technical aspects with certain in-depth details.  Otherwise, this is a good written paper with clear description of methods, results and discussions. If authors can add the detailed description of Ultrasound shear wave detection and how the ultrasound modulus was obtained, this paper can be accepted.  

Author Response

Dear Reviewer,

We would like to thank you for appreciating our paper and its contribution to the diagnosis of carpal tunnel syndrome. Thank you for carefully reading our manuscript. We are glad you found it interesting. We thank you for giving us useful suggestions to improve our article. Changes made to the text are highlighted in yellow. Below are our responses to your suggestions.

First reviewer's comments:

The written of this manuscript is quite well. However, from scientific point of view, the description of the technical methods is not very clear. The manuscript needs to describe how the ultrasound shear wave was measured from ultrasonic technical aspects with certain in-depth details.  Otherwise, this is a good written paper with clear description of methods, results and discussions. If authors can add the detailed description of Ultrasound shear wave detection and how the ultrasound modulus was obtained, this paper can be accepted. 

Thank you for your valuable feedback on our manuscript. We have carefully considered your suggestions and have significantly revised the Material and methods chapter, subsection Ultrasound measurements.

We have addend:

US imageswere collected using SWE mode installed in the US machine. Immageswere collected in 5 different positions of both radiocarpal joints (intermediate position 0°, 45° of extension, maximum extension, 45° of flexion, and maximum flexion).

and

Three images and then three measurements were taken at each angular position and the average values from these measurements were used for the final analysis. For both CSA and shear modulus measurements, a contour was made of the median nerve at the inner hypoechoic border using a Q-Box Trace option implemented in the US machine (Figure 2). Both parameterswere automatically calculated in US machine and expressed in square millimetres and,  kilopascals (kPa) for CSA and shear modulus, respectively.CSA is related to nerve surface area, whereasshear modulusreferes to stiffness of the nerve.Higher values of shear modulus indicate higher nerve stiffness and vice versa. Detailed information about the basic physics and musculoskeletal applications of SWE is placed elsewhere [35].

Reviewer 2 Report

Comments and Suggestions for Authors

The research is about the findings and the potential implications for the diagnosis and treatment of Carpal Tunnel Syndrome (CTS).

1- The literature review has explained many works, but there is no detailed explanation of recent literature (especially the last 5 years). Instead of using old techniques, you must compare your method and benefits with the recent methods.

2- the dataset is a little bit small and specifically focuses on one type of patient (Polish).  This does not allow you to make a generalized approach. The dataset must be large and diverse to have more robust results.

3- You can add doi numbers in the references.

Overall structure of paper is good. Objective is clear and contribute valuable insights into use of ultrasound in diagnosing CTS, but remember if you can cover '2nd' suggestion.

Besides, you have mentioned in the limitation part to include nerve condition studies for asymptomatic limb.  If you can find data about them, it improves your work.

Author Response

Second reviewer's comments:

1- The literature review has explained many works, but there is no detailed explanation of recent literature (especially the last 5 years). Instead of using old techniques, you must compare your method and benefits with the recent methods.

Thank you for your suggestions. We have expanded the literature with the latest reports. In the discussion part of the paper, we added an entire paragraph presenting the latest work.

We have addend:

  1. El-Maghraby AM, Almalki YE, Basha MAA, Nada MG, El Ahwany F, Alduraibi SK, Alshehri SHS, Aldhilan AS, Almushayti ZA, Alduraibi AK, Aboualkheir M, Attia O, Amer MM, Basha AMA, Eladl IM. Diagnostic Accuracy of Integrating Ultrasound and Shear Wave Elastography in Assessing Carpal Tunnel Syndrome Severity: a Prospective Observational Study. Orthop Res Rev. 2024:16:111-123. Doi: 10.2147/ORR.S459993. eCollection 2024.
  2. Martikkala L, Pemmari A, Himanen SL, Mäkelä K. Median Nerve Shear Wave Elastography Is Associated With the Neurophysiological Severity of Carpal Tunnel Syndrome. J Ultrasound Med. 2024;43(7):1253-1263.Doi: 10.1002/jum.16450.
  3. Prakash A, Vinutha H, Janardhan DC, Mouna RM, Sushmitha PS, Sajjan S, Samanvitha H. Diagnostic efficacy of high-frequency Grey-scale ultrasonography and Sono-elastography in grading the severity of carpal tunnel syndrome in comparison to nerve conduction studiem. Skeletal Radiol. 2024 Doi: 10.1007/s00256-024-04662-y.
  4. Khademi S, Kordi Yoosefinejad A, Motealleh A, Rezaei I, Abbasi L, Jalli R. The sono-elastography evaluation of the immediate effects of neurodynamic mobilization technique on median nerve stiffness in patients with carpal tunnel syndrome. J Bodyw Mov Ther. 2023:36:62-68.Doi: 10.1016/j.jbmt.2023.01.001.
  5. Wu H, Zhao HJ, Xue WL, Wang YC, Zhang WY, Wang XL. Ultrasound and elastography role in pre- and post-operative evaluation of median neuropathy in patients with carpal tunnel syndrome. Front Neurol. 2022:13:1079737. Doi: 10.3389/fneur.2022.1079737.

Recently, reports on the possibility of using US and SWE in the diagnosis of CTS are increasingly appearing in the literature. El-Maghraby et al. [47] assessed the usefulness of combined diagnostic utility of B-mode US and SWE for assessing the severity of CTS in comparison to electrodiagnostic tests. Based on the obtained research results, the authors concluded that the integration of US, SWE, and electrodiagnostic tests provides a comprehensive approach to evaluate anatomical and neurological changes and guide management decisions for CTS. Similar conclusions were drawn from their research by Martikkala et al. [48] ​​indicating that shear wave velocity in the median nerve at the carpal tunnel inlet increases in CTS and correlates to the neurophysiological CTS severity equivalently to CSA measured at the same site. In turn, Prakash et al. [49] assessed the diagnostic effectiveness of high-frequency US and SWE in grading the severity of CTS. The authors conclude that the combined utility of US and SWE may serve as a painless and cost-effective alternative to nerve conduction study in grading the severity of CTS. Interesting research was conducted by Khademi et al. [51] assessing SWE and CSA of the median nerve in people with CTS after using neurodynamic techniques. Researchers have shown that one session of neurodynamic techniques reduced the stiffness and CSA of median nerve at wrist in patients with CTS. The authors emphasize that this confirms that SWE is able to quickly detect the immediate biomechanical changes of the median nerve. Wu et al. [52] showed the usefulness of conventional US and SWE in the clinical assessment of patient's nerve recovery after surgery. We believe that performing such an examination in different angular positions may better reflect changes in the morphology of the nerve. Our report expands the topic of the use of US and SWE in the diagnosis of CTS, indicating that it is beneficial to examine the median nerve (CSA and shear modulus) not only in the neutral position of the wrist, but also in other angular positions. This will be important not only for more detailed diagnostics, but also for a better assessment of the final results of the therapy used.

2- the dataset is a little bit small and specifically focuses on one type of patient (Polish).  This does not allow you to make a generalized approach. The dataset must be large and diverse to have more robust results.

Thank you for this suggestion. I agree with the respected reviewer, which is why we added it as a limitation of this work.

We have addend:

The first limitation is the relatively small group of CTS patients studied and specifically focuses on one type of patient (Polish). This does not allow  to make a generalized approach to the general population. 

3- You can add doi numbers in the references.

Thank you for your suggestion. We have added DOI numbers.

Overall structure of paper is good. Objective is clear and contribute valuable insights into use of ultrasound in diagnosing CTS, but remember if you can cover '2nd' suggestion.

Thank you for appreciating our work.

Besides, you have mentioned in the limitation part to include nerve condition studies for asymptomatic limb.  If you can find data about them, it improves your work.

Thank you for this insight. Unfortunately, we have not found such studies so far.